**Ozone Comparison between Pandora #34,  Dobson #061, OMI, and OMPS at Boulder Colorado for the period December 2013 – December 2016.**

**Jay. Herman[1], Robert Evans[4], Alexander Cede[3], Nader  Abuhassan[1], Irina. Petropavlovskikh[2], Glenn McConville[2], Koji Miyagawa[5], and Brandon Noirot[2]**

[1] University of Maryland Baltimore County (JCET) at Goddard Space Flight Center, Greenbelt, MD

[2] NOAA Earth System Research Laboratory, Boulder, CO. Cooperative Institute for Research in Environmental Sciences (CIRES), University of Colorado, Boulder, CO

[3] LuftBlick, Austria and Goddard Space Flight Center, Greenbelt, MD

[4]Former scientist at NOAA/ESRL/GMD, Boulder, CO:  Retired

[5]Guest Scientist at NOAA/ESRL/GMD, Boulder, CO

**Abstract**

A one-time calibrated (in December 2013) Pandora Spectrometer Instrument (Pan #034) has been compared to a periodically calibrated Dobson spectroradiometer (Dobson #061) co-located in Boulder, Colorado, and compared with two satellite instruments over a 3-year period. The results show good agreement between Pan#034 and Dobson#061 within their statistical uncertainties. Both records are corrected for ozone retrieval sensitivity to stratospheric temperature variability obtained from the Global Modeling Initiative (GMI) and Modern-Era Retrospective analysis for Research and Applications (MERRA-2) model calculations. Pandora#034 and Dobson#061 differ by an average of 2.1 ± 3.2 % when both instruments use their standard ozone absorption cross sections in the retrievals algorithms. The results show a relative drift (0.2 ± 0.08% per year) between Pandora observations against NOAA Dobson in Boulder, CO over a three-year period of continuous operation.  Pandora drifts relative to the satellite Ozone Monitoring Instrument OMI and the Ozone Mapping Profiler OMPS are +0.18 ± 0.2 % per year and -0.18 ± 0.2 % per year, respectively, where the uncertainties are 2 standard deviations. The drift between Dobson #061 and OMPS for a 5.5-year period (January 2012 – June 2017) is -0.07 ± 0.06 % per year.

Author(s): Jay Herman et al.
MS No.: amt-2017-157
MS Type: Research article
Iteration: Revised
Special Issue: Quadrennial Ozone Symposium 2016  Status and trends of atmospheric ozone (ACP/AMT inter-journal SI)

## Introduction

A Pandora Spectrometer Instrument #034 (PSI) located on top of the NOAA building in Boulder, Colorado has been operating since December 2013 with little maintenance and using the original calibration. The purpose of this paper is to present a comparison between two co-located ozone measuring instruments, Pandora #034 and Dobson #061 for the period December 2013 to December 2016. Additional comparisons are made with satellite overpass data from OMI (Ozone Monitoring Instrument on board the AURA spacecraft) and OMPS (Ozone Mapping Profiler Suite on board the Suomi NPOESS satellite). This paper is an extension of a previously published paper (Herman et al., 2015) that presented just 1 year of data. The results demonstrate the accuracy and stability of both the Dobson and PSI for retrieval of total column ozone, and serves as a validation demonstration at one location for both the fairly new PSI and for satellite ozone data from OMI and OMPS. Part of the experiment comparing Pandora #034 to Dobson #061 was to see if Pandora #034 would perform well over a long period without additional calibration or adjustments. The only change made during the period 2014 to the present (August 2017) was to replace a broken motor on the suntracker that caused a data gap in early 2016.

The characteristics of both the PSI and the Dobson Spectroradiometer are described in Herman et al. (2015). Briefly, the PSI consists of a small Avantes low stray light spectrometer (280 – 525 nm with 0.6 nm spectral resolution with 5 times oversampling) connected to an optical head by a 400 micron core diameter single strand fiber optic cable. The spectrometer is temperature stabilized at $20^O$C inside of a weather resistant container. The optical head consists of a collimator and lens giving rise to a $2.5^O$ FOV (field of view) FWHM (Full Width Half Maximum) with light passing through two filter wheels containing diffusers, open hole, a UV340 filter (blocks visible light), neutral density filters, and an opaque position (dark current measurement). The optical head is connected to a small suntracker capable of accurately following the sun's center using a small computer-data logger contained in a weatherproof box along with the spectrometer. Pandora#034 is capable of obtaining $NO_2$ and Total Column Ozone TCO amounts sequentially over a period of 80 seconds. The integration time in bright sun is about 4 milli-seconds that is repeated and averaged for 30 seconds to obtain very high signal to noise and an ozone precision of less than 1 DU or 0.2% (1 DU = $2.69 \times 10^{16}$ molecules/cm$^2$).

The Dobson record in Boulder started in 1966 based on an improved design from the instrument first deployed in the 1920's (Dobson, 1931). Dobson instrument is using differential absorption method to derive total column ozone from direct–sun measurements using two UV wavelength pairs in the 300 – 340 nm range (see Herman et al., 2015). The extensive Dobson network uses the Bass-Paur (BP) ozone absorption cross sections (Bass and Paur, 1985) for operational data processing (Komhyr et al., 1993).

All NOAA Dobson instruments are periodically calibrated against WMO world standard Dobson #083, which in turn uses Langley method calibrations at the Mauna Loa Observatory station (Komhyr et al., 1989). Standard lamps are used to check Dobson spectral registration stability. Recently, July 2017, intermediate calibrations were applied to the Dobson #061 ozone data record that improved its comparison with satellite data (the calibration updates were processed by one of the co-authors, Koji Miyagawa).

The main sources of noise in the PSI measurement comes from the presence of clouds or haze in the FOV, which increases the exposure time needed to fill the CCD wells to 80% and reduces the number of measurements in 20 seconds. For this comparison study, data were selected for scenes that are clear-sky conditions as determined from the Dobson A-D pair direct-sun data record.

Accuracy in the PSI spectral fitting retrieval is obtained using careful measurements of the spectrometer's slit function, wavelength calibration, and knowledge of the solar spectrum at the top of the atmosphere. The current operational PSI ozone retrieval algorithm used in this study is based on extraterrestrial solar flux from a combination of the Kurucz spectrum (wavelength resolution $\lambda/1\lambda$ = 500 000) radiometrically normalized to the lower-resolution shuttle Atlas-3 SUSIM spectrum (Van Hoosier, 1996; Bernhard et al., 2004, 2005), BDM ozone cross sections (Brion et al. (1993, 1998) and Malicet et al. (1995)), corrections for stray light, and an effective ozone weighted temperature.

The Dobson data used in this study contain the individual measurements (more than 1 per day between 09:00 and 15:00 local time with almost all of the data between 10:00 and 14:00) for clear-sky direct-sun observations using the quartz plate and A-D wavelength pairs for ozone retrieval. These were made available by one of the co-authors (I. Petropavlovskikh, private communication, Table 2). The NOAA Dobson total ozone data are typically archived at WOUDC (World Ozone and Ultraviolet Radiation Data Centre) or NDACC (Network for the Detection of Atmospheric Composition Change) with one representative ozone value per day**.**

1. **Temperature Sensitivity**

The PSI ozone retrieval algorithm is more sensitive to the effective ozone weighted average temperature than is the 4 wavelength Dobson retrieval (Redondas et al., 2014). Neglecting the temperature sensitivity creates a seasonal difference between the two instruments. To correct for this, we use an effective ozone temperature $T_E$ based on daily ozone weighted altitude temperature averages (Redondas et al., 2014). The temperature and ozone profile data were obtained from the GMI (Global Modeling Initiative) model calculation for 2012 to 2016. (https://gmi.gsfc.nasa.gov/merra2hindcast/). The GMI model provides atmospheric composition hindcasts using MERRA-2 (Modern-Era Retrospective analysis for Research and Applications,

Version 2, meteorology (Strahan et al., 2013; Wargan and Coy, 2012)
https://gmao.gsfc.nasa.gov/reanalysis/MERRA-2/). The simulation with 2 x 2.5 degree resolution
uses the CCMI (Chemistry–Climate Modelling Initiative, Morgenstern et al., 2017) emissions
and boundary conditions. MERRA-2 uses assimilation schemes based on hyperspectral radiation,
microwave observations and ozone satellite measurements. The resulting seasonal cycle for $T_E$
shows variations over the four year period, while day-to-day variability is enhanced during
winter and spring season (Fig. 1). An estimated $5^{th}$ year (2017) has been added (Fig. 1) by
forming the average of the daily temperatures from the 2013 to 2016 period.

The $T_E$ time series data are used for an ozone retrieval temperature correction $TCO_{cor}$

coefficient per $^O K$ given in the form $TCO_{corr} = TCO (1 + C(T))$ and $O_3(corr) = O_3\ TCO_{corr}$
(Herman et al., 2015), where $C(T_E)$ is given by eqns. 1 and 2.

    $C_{Pandora-BDM}(T_E) = 0.00333(T_E - 225)$        (Herman et al., 2015)        (1)

    $C_{Dobson-BP}(T_E) = -0.0013(T_E - 226.7)$       (Redondas et al., 2014)     (2)

    $C_{Dobson-BDM}(T_E) = 0.00042(T_E - 226.7)$     (Redondas et al., 2014)     (3)

As mentioned earlier, the Dobson TCO retrieval normally uses the Bass and Paur (BP)

ozone absorption coefficients, while Pandora uses the Brion-Daumont-Malicet (BDM)
coefficients. A change in $T_E$ of $+1^O$ change leads to TCO changes for the Pandora(BDM),
Dobson(BP), and Dobson(BDM) instruments of +0.33%, -0.13%, and 0.042%, respectively.
For a nominal TCO value of 325 DU, the change would be +1.1 and -0.4 DU, a net relative
change of 1.5 DU for a $1^O K$ change between Pandora(BDM) and Dobson(BP).

While BDM cross sections are not currently recommended for use in standard Dobson

processing, their use yields slightly different values of TCO and a smaller sensitivity to
temperature. The basic Dobson algorithm, based on pairs of wavelengths, is intrinsically less
sensitive to $T_E$ than Pandora's spectral fitting retrieval.

**2.  TCO Comparisons between Pandora, Dobson, OMI and OMPS**

Comparing retrieved TCO from the PSI, Dobson, OMI and OMPS instruments show that

there are small, but significant differences between the PSI and Dobson instruments and between
the ground-based instruments and satellite derived values of TCO.  The difference is calculated
using three-year estimates of secular change based on a linear least squares fit to the percent
differences between the instruments. The cloud-free direct-sun A-D pair Dobson ozone data are
selected for comparison with time-matched Pandora#034 retrieved ozone data (Herman et al.,
2015). The Pandora#034 retrieved ozone (every 80 seconds) are matched to the less frequent
Dobson#061 retrieval times that are obtained for mid-day solar zenith angles (SZAs) and
averaged over ±8 minutes (Fig. 2A).

Each clear-sky PSI data point is an average of 2000 (early morning to evening SZAs) to
4000 (mid-day SZAs) measurements obtained during 20 seconds. All data for this study were
clear-sky within the instrument's field of view based on the Dobson criteria for A-D-pair direct-
sun clear sky.  In addition, the PSI data are averaged over a period of +/- 8 minutes surrounding
the Dobson time of measurement (2 to 3 times per day). Since PSI measurements are obtained
every 80 seconds, there were an additional 10 PSI data points averaged together to compare to
each Dobson, OMI, or OMPS measurement. The result is high signal to noise values for Pandora
and high precision (0.1%). The same procedure using cloud-screened PSI data was used for
comparisons with OMI and OMPS, where they measure once or twice per day over Boulder,
Colorado. Some of the variations in the day to day ozone values are driven by changes in the
local weather over Boulder, Colorado (see Fig. 14 in Herman et al., 2015), with weekly averages
having much smaller variation.

Figure 2B shows a Lowess(0.1) fits to the two time series in Fig. 2A that is approximately
equivalent to a 3-month running average. The Lowess(f) procedure is based on local least
squares fitting using low order polynomials applied to a specified fraction f of the data
(Cleveland, 1979) that reduces the effect of outlier points from the mean. The smooth curves
show a small variable difference between the Dobson and Pandora time series. Fig. 2C shows the
percent difference PD between the time series in Fig. 2A and the residual seasonal variation in
PD. Estimating the slope of the least squares fit to the percent difference is sensitive to the
selection of the end points of the time series. This effect can be minimized by removing the
seasonal time dependence (Fig. 2C) using a low-pass filter function with zero slope derived from
the Lowess(0.1) fit. The result is shown in Fig. 3A.

Figure 3 shows the de-seasonalized percent differences PD(A,B) for six pairs between
Pandora #034, Dobson #061, OMI, and OMPS for the 3-year period 2014 – 2016 (summarized in
Table 1).  The slightly curvy Lowess (0.1) lines about each linear fit show the residual seasonal
cycles, which are too small to have an effect on slope determination. Error estimates (Fig. 3 and
Table 1) for the linear least squares slopes and averages are one standard deviation (1-STD).
Some of the error estimates are large enough to make the statistical significance of the slopes
marginal (see Panel E OMPS vs Pandora; $0.18 \pm 0.098$, $p = 0.06$), while others are significant
(see Panel D OMI vs Dobson: $-0.18 \pm 0.08$,  $p = 0.03$) at the 2-STD level. The significance
probability parameter p is given, where p is the probability (0 to 1) that the slope is statistically
different from 0 relative to $p = 0.05$. Also shown are the numbers of data points in each time
series.

After removal of the residual seasonal variation in the calculated percent differences,
there still is a statistically significant drift of 0.2% per year ($p < 0.001$) between the Pandora#034
and Dobson#061 (Panels A and B in Fig. 3) using either BP or BDM ozone cross sections for the
Dobson. The differences in the mean values (-2.1 and -2.8%) are not significant at the 2-STD
level.
The linear trend (Panel C, -0.09 ± 0.08 % per year, p = 0.3) between the Dobson and
OMPS is not significantly different from zero, while the drift with OMI (Panel D,-0.18 ± 0.08 %
per year, p = 0.03) is significant. This suggests that OMI ozone retrievals are drifting with
respect to OMPS and the Dobson. Extending the period from 2012 to June 2017 gives a very
small, but significant trend, -0.07 ± 0.03 % per year, p = 0.047 for PD(OMPS,Dobson).
Calculations for Pandora#034 (Panels E and F in Fig. 3) show marginally significant (p =
0.06) trends for Pandora#034 compared to OMPS (Panel E, -0.18 ± 0.098 % per year) and OMI
(Panel F, +0.18 ± 0.096 % per year).  If the Pandora#034 time series is extended into 2017 to
minimize the effect of missing Pandora data in 2016, then the trends for Pandora compared to
OMPS (-0.2 ± 0.08 % / Year   p = 0.013) and OMI (0.15 ± 0.076 p=0.05) are significant, but not
different from the shorter 2014 – 2016 period. The secular trends for the difference between
Pandora#034 and Dobson#061 (-0.2% per year) are almost the same for both Dobson BP and
BDM ozone absorption coefficients even though the temperature sensitivity using the Dobson
BDM ozone absorption coefficients is small (0.042% per $^O$C). This suggests that the
stratospheric effective ozone temperature change is not a source for the small difference between
Pandora#034 and Dobson#061.
Figure 4 shows that the TCO between Pandora#034 and Dobson#061 are highly
correlated with 1:1 slope and the correlation coefficient $r^2$ = 0.97 for the 3-year period 2014 to
2016. Similar correlation plots (Fig. 5) for Pandora#034 and Dobson#061 with OMI and OMPS
also show very high correlations. The correlations in TCO are obtained after only temperature
corrections to Pandora#034 and Dobson#061 using $T_E$ (TCO pairs similar to Fig. 2, panel A).
The Pandora, OMI, and OMPS data used in this study are from the overpass files located
on the public websites (Table 2).

**Summary**
Temperature corrected Pandora#034 and Dobson#061 differ by an average of 2.1% with
Pandora using its standard retrieval BDM ozone absorption cross sections and Dobson using the
recommended BP ozone absorption cross sections. Pandora compared to Dobson shows a small,
but significant drift (-0.2 ± 0.04 % per year,  p < 0.001) for the 2014 – 2016 period. Comparisons
of Pandora with OMI and OMPS are marginally significant drifts of 0.18±0.1 and -0.18±0.1
p=0.06 for 2014-2016, but are significant (0.15 ± 0.076 % per year, p=0.05 and -0.2 ± 0.08 % per
year,   p = 0.013, respectively) if the period is extended to mid-2017 to minimize the effect of
missing Pandora data during 2016. The small Pandora and Dobson trends compared to OMPS
suggests that both instruments are stable. The conclusion is that the periodically calibrated

Dobson#061 is able to detect smaller ozone trends than a Pandora instrument with no intermediate calibration during a 3-year period. The longer term trend for Dobson compared to OMPS for a 5.5-year period (2012 – June 2017) is -0.07 ± 0.03 % per year, p = 0.047.

**Acknowledgement:** The authors would like to thank Dr. Susan Strahan and the MERRA-2 team for supplying the atmospheric temperature data for Boulder, Colorado.

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

## Tables

Table 1 Percent Difference Summary of Linear Fit Slopes and Mean Differences in Fig. 3

| Percent Diff(A,B) | Slope (% per Year) | Probability | Mean (%) | Points | Panel |
|---|---|---|---|---|---|
| Pan, Dob(BP) | -0.2 ± 0.04 | P < 0.001 | -2.1 ± 1.6 | 2020 | A |
| Pan, Dob(BDM) | -0.2 ± 0.04 | P < 0.001 | -2.8 ± 1.6 | 2020 | B |
| OMPS, Dob(BP) | -0.09 ± 0.08 | P = 0.3 | -1.4 ± 2.1 | 854 | C |
| OMI, Dob(BP) | -0.18 ± 0.08 | P = 0.03 | -1.4 ± 1.9 | 654 | D |
| OMPS, Pan | -0.18 ± 0.098 | P = 0.06 | 0.96 ± 2.7 | 952 | E |
| OMI, Pan | +0.18 ± 0.096 | P = 0.06 | 1.1 ± 2.1 | 624 | F |

Table 2 Data Availability

**OMI:**
https://avdc.gsfc.nasa.gov/index.php?site=1593048672&id=28/aura_omi_l2ovp_omto3_v8.5_boulder.co_067.txt
**OMPS**:
ftp://toms.gsfc.nasa.gov/pub/omps_tc/overpass/suomi_npp_omps_l2ovp_nmto3_v02_boulder.co_067.txt
**Pandora34:**
https://avdc.gsfc.nasa.gov/pub/DSCOVR/Pandora/DATA/Boulder/Pandora34/L3c/

**Dobson061**:
ftp://aftp.cmdl.noaa.gov/data/ozwv/Dobson/WinDobson/Pandora%20comparisons/Dobson61%20Boulder%20Ad-dsgqp%2020120213-032717_w_Header.txt

**Figure Captions**

Fig. 1 Calculated $T_E$ using model estimates of $O_3$ and temperature profiles. The Trend is calculated from the difference of $T_E$ from its 4-year daily mean that is also used for year 2017 labelled Avg.

Fig. 2  Panel A shows the retrieved ozone time series (December 2013 – June 2017) for Pandora (red) and Dobson (Black). Panel B shows Lowess(0.1) fit to the each time series.  Panel C shows the percent difference, a linear least squares fit, and a Lowess(0.1) fit showing seasonal residuals.

Fig. 3 Comparisons of Pandora(BDM) with Dobson(BP) retrieved ozone for Boulder, Colorado in percent differences of retrieved ozone and comparisons with OMI and OMPS. Slope = value of the linear least square fit, ±N is 1 STD, and p is the probability (0 to 1) that the slope is statistically different from 0 relative to p = 0.05. The solid lines are a Lowess(0.1) fit and a linear least squares fit.

Fig. 4 Correlation between Pandora #034 and Dobson #061: 2014 – 2016

Fig. 5 Correlation of Pandora#034 and Dobson#061 with OMI and OMPS: 2014 - 2016

**Figures**

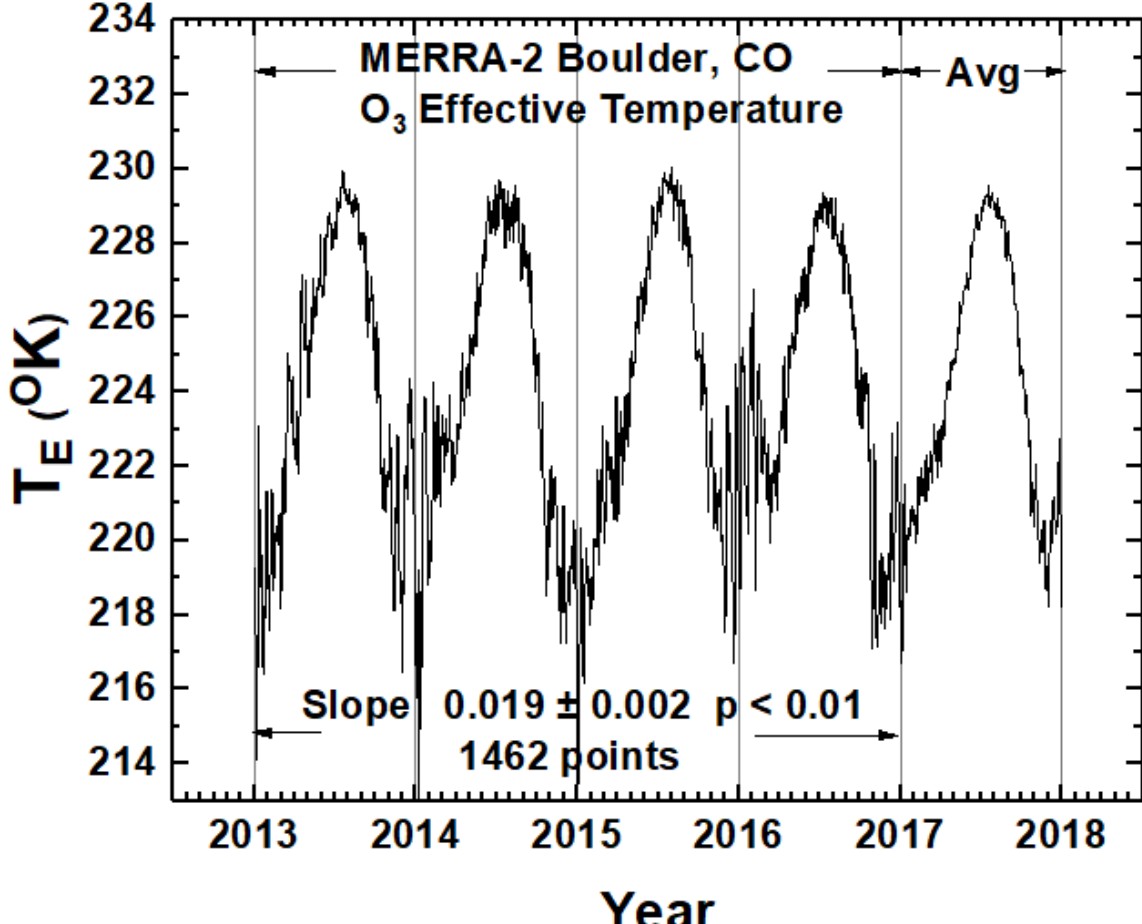


**F1**

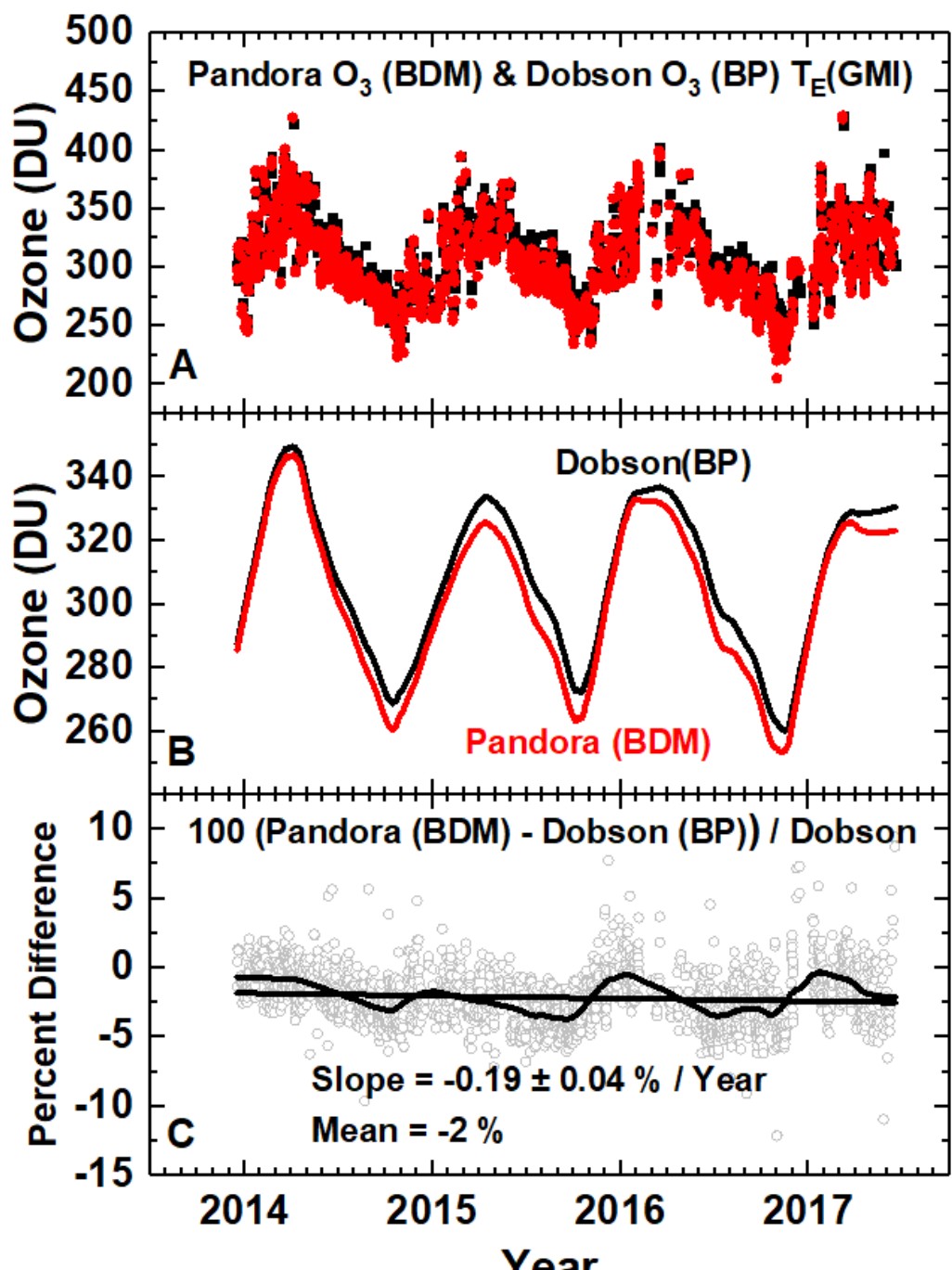

**F2**

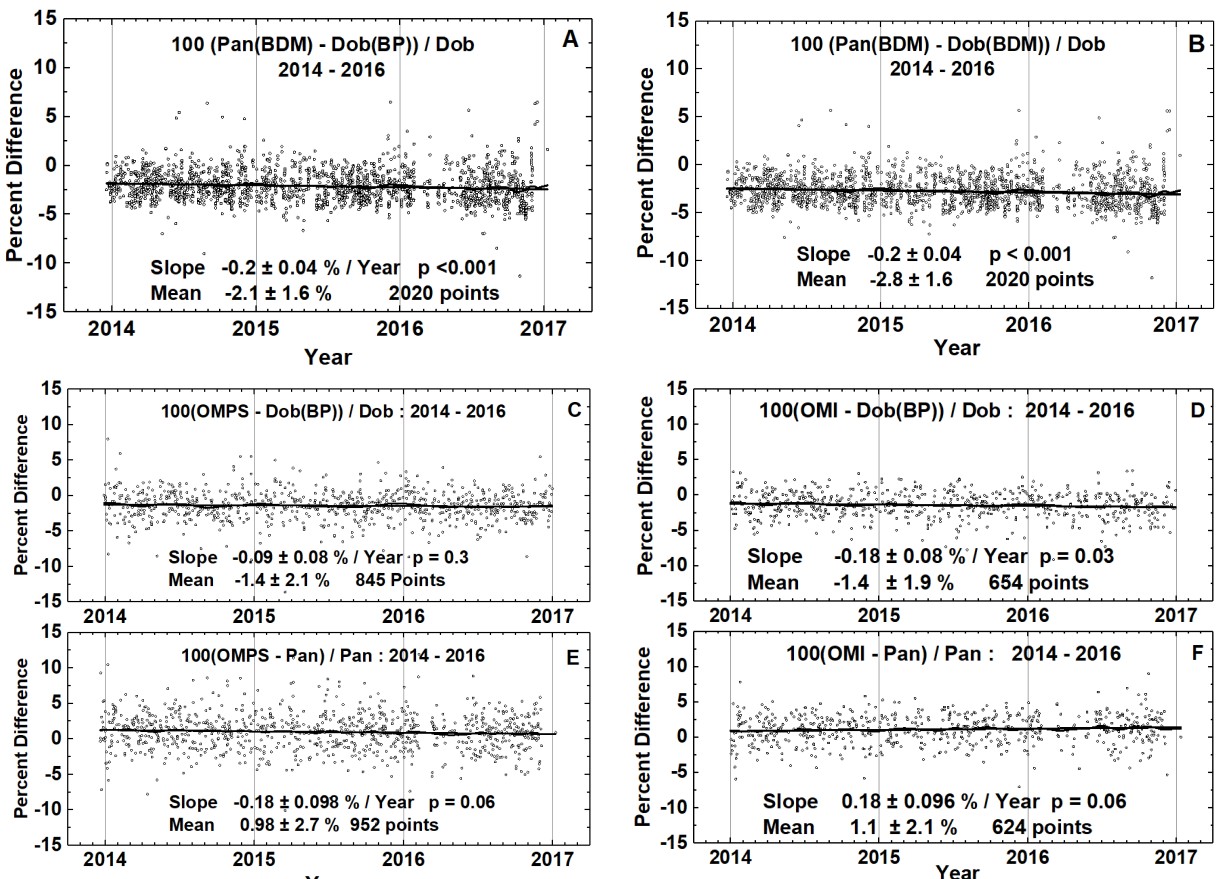



**F3**


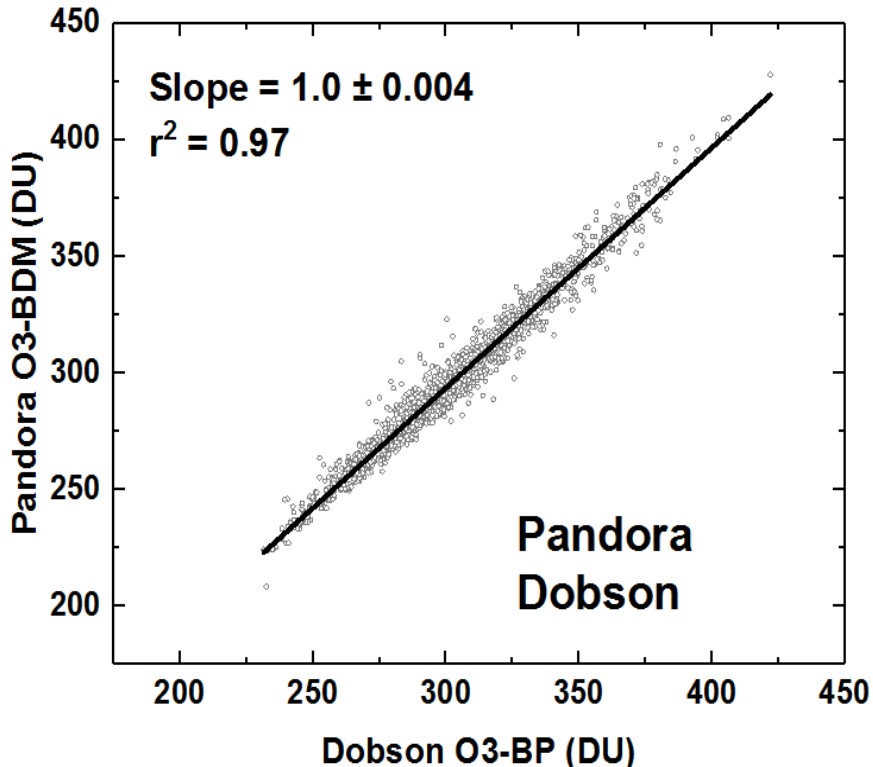

**F4**

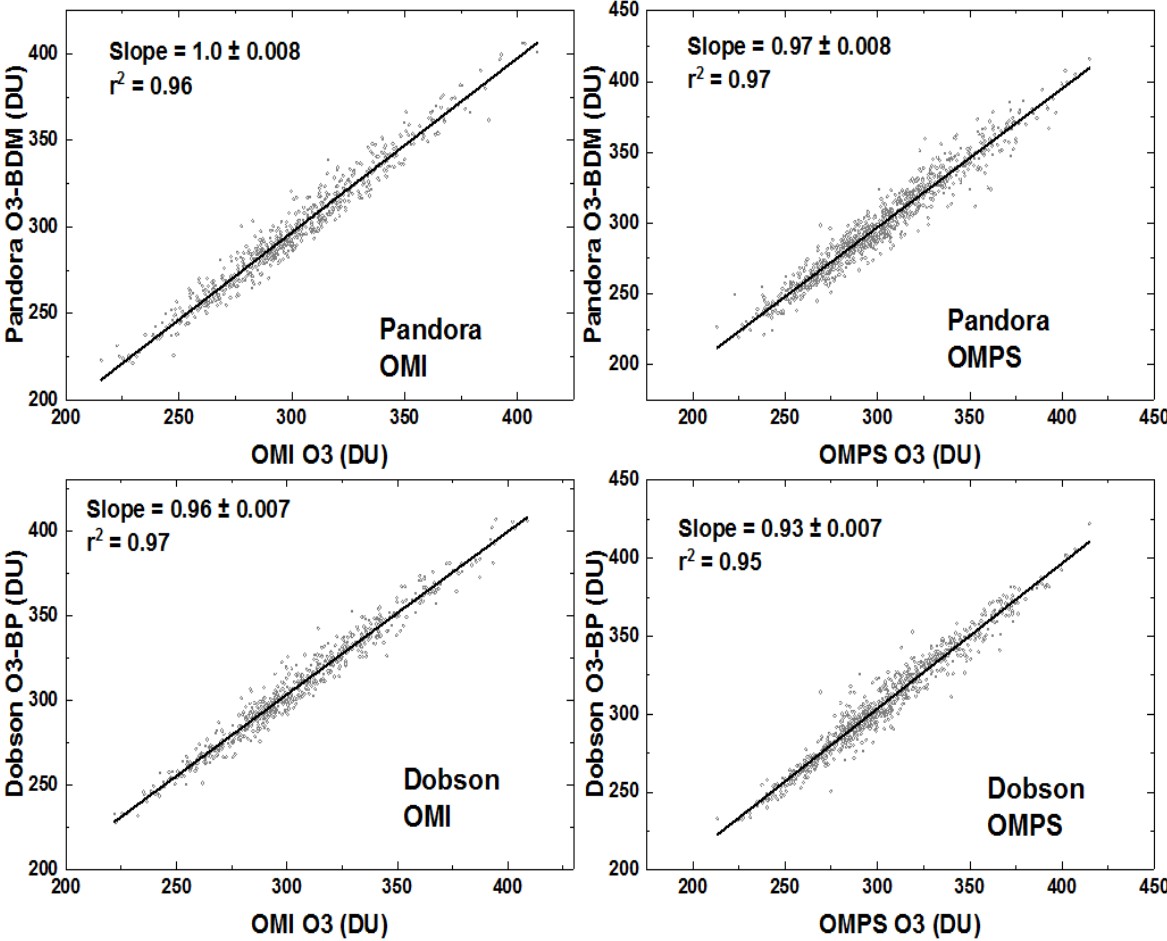

**F5**