# Peer review of "Ozone Comparison between Pandora #34, Dobson #061, OMI, and OMPS at Boulder"

_Atmospheric Measurement Techniques, 2017_

## Referee Comment (RC1) · R. Chatfield (Referee) · 30 Jun 2017

This is a good basic publication which just needs clarity and precision. Conclusions regarding trends in retrieved ozone need more modest standard error estimates, I believe

The authors appear to make two assumptions:

[Figure]

(a) That "significance" means a 5% (? not stated) chance of Type 1 error (false acceptance) with a Gaussian distribution of errors.

(b) That the "number of relevant samples" is the number of individual observations, apparently as averaged for 80 seconds for the PANDORA, the number of individual observations (averaged over 8 minutes, or once daily?) of observation recorded for the Dobson, and the number of days of observation (maximum once per day?) for OMI and OMPS. For some comparisons, "data were selected for scenes that are clear-sky conditions as determined from the Dobson A pair" For all? How many days? Each of these numbers should be stated in the relevant context . There are many statistics quoted where the reviewer was confused. Please describe each.

The appropriate statistic to quote is the p-value (0.05 ??) with the number of observations used in each statistic, and one- or two-sided calculation, where there could be confusion. For example, a p–value of 0.10 would suggest to the reviewer that there was something worth further investigation.

The point of maximum confusion for the reviewer was the discussion of drift. What number of samples was used? The eye sees that "independent" observations seem to occur often due to some rapidly changing condition: experimental error in one or both instruments, or rapid weather variation? The smoothed lines (which smoothing for Figure 3 as Figure 2. lowess(0.1), reference, explain "0.1)"?) suggest that "weather" variation has a substantial impact on the smoothes and indeed the trends, especially in Figure 2. The smoothes for Figure 2 appear somewhat more convincing, but the uncertainty of 0.1% seems to be based on number of all samples rather than some partial contribution from "weather variability."

One could guess a synoptic value of "five days per synoptic episode" and calculate a debatable approximate "number of samples" but the more appropriate value would be derived from a time series analysis which allowed for longer time-scales in that algorithm.

In fact, there is enough excellent data here for most series to justify a more careful time-series analysis. For this publication, a disclaimer saying that "weather variability" could allow for a larger uncertainty in the apparent divergence is acceptable. In this case, "weather" is longer than one day but probably shorter than three years. Similar comments apply to the +/- 0.002 in Figure 1.

(minor points: explain acronym CCMI; perhaps OMI and OMPS are named on web pages, but could explained)

This will be a nice addition to the description of stratospheric (and tropospheric) change and tropospheric change (TOAR). We may hope that the advent of many PANDORA instruments will add to a better discrimination of the variability and secular change of ozone as a function of altitude. Minimal re-review is expected.

1. Does the paper address relevant scientific questions within the scope of AMT? Yes 2. Does the paper present novel concepts, ideas, tools, or data? Yes, Data 3. Are substantial conclusions reached? Yes, sufficient when they are qualified as noted 4. Are the scientific methods and assumptions valid and clearly outlined? Correctable. See notes above 5. Are the results sufficient to support the interpretations and conclusions? Ditto 6. Is the description of experiments and calculations sufficiently complete and precise to allow their reproduction by fellow scientists (traceability of results)? Ditto 7. Do the authors give proper credit to related work and clearly indicate their own new/original contribution? Yes 8. Does the title clearly reflect the contents of the paper? Yes 9. Does the abstract provide a concise and complete summary? Yes 10. Is the overall presentation well structured and clear? Yes 11. Is the language fluent and precise? Yes, but see 4. 12. Are mathematical formulae, symbols, abbreviations, and units correctly defined and used? Yes, minor additions needed for abbreviations, see above for e.g. "significant" and "Lowess(0.1)" 13. Should any parts of the paper (text, formulae, figures, tables) be clarified, reduced, combined, or eliminated? No 14. Are the number and quality of references appropriate? Yes 15. Is the amount and quality of supplementary material appropriate? Yes

---

## Author Comment (AC1) · 5 Jul 2017

AMTD Interactive comment Printer-friendly version Discussion paper Atmos. Meas. Tech. Discuss., doi:10.5194/amt-2017-157-RC1, 2017 © Author(s) 2017. This work is distributed under the Creative Commons Attribution 3.0 License. Interactive comment on "Ozone Comparison between Pandora #34, Dobson #061, OMI, and OMPS at Boulder Colorado for the period December 2013–December 2016" by Jay Herman et al. R. Chatfield (Referee) Robert.B.Chatfield@nasa.gov Review of Herman et al.

Reply: I will incorporate the replies into the paper after the second review (if any).

This is a good basic publication which just needs clarity and precision. Conclusions regarding trends in retrieved ozone need more modest standard error estimates, I believe The authors appear to make two assumptions: (a) That "significance" means a 5% (? not stated) chance of Type 1 error (false acceptance) with a Gaussian distribution of errors.

Reply:The error estimates given are 1 standard deviation STD estimated from the least squares linear fit process. The error of the individual points was described in the previous paper as 1% with a precision of 0.1%. Most of the variation seen in the data is "natural" variation". It is clear that the error bars are large enough to make the statistical significance of some of the slopes marginal (see OMPS vs Pandora; 0.19 +/- 0.1). Some of the others are significant (see OMPS vs Dobson: -0.4 +/- 0.09) at the 2 STD level.

(b) That the "number of relevant samples" is the number of individual observations, apparently as averaged for 80 seconds for the PANDORA, the number of individual observations (averaged over 8 minutes, or once daily?) of observation recorded for the Dobson, and the number of days of observation (maximum once per day?) for OMI and OMPS.

Reply: Each Pandora data point is an average of 4000 measurements obtained during 20 seconds. All data for this study were clear-sky within the instrument's field of view based on the Dobson criteria for A-pair direct-sun clear sky. In addition, the Pandora data are averaged over a period of +/- 8 minutes surrounding the Dobson time of measurement (2 to 3 times per day). Pandora measurements are obtained every 80 seconds that means there were an additional 10 Pandora data points averaged together to compare to each Dobson measurement. The net averaging of Pandora is 40,000 (4x104) measurements for each comparison. The same procedure was used for comparisons with OMI and OMPS, where they measure once or twice per day over

Boulder, Colorado.

For some comparisons, "data were selected for scenes that are clear-sky conditions as determined from the Dobson A pair" For all?

Reply: All Dobson vs Pandora, OMI, or OMPS scenes were clear-sky A-pair using the Dobson criterion Reply: All Pandora vs OMI and OMPS were clear sky using the Pandora criterion   How many days? Each of these numbers should be stated in the relevant context .

Reply: Dobson vs Pandora has 1326 points with 1 to 3 points per day OMI vs Pandora has 637 points with 1 -2 points per day OMPS vs Pandora has 956 points with 1 – 2 points per day OMI vs Dobson has 636 points with 1 – 2 points per day OMPS vs DOBSON has 833 points with 1 - 2 points per day

There are many statistics quoted where the reviewer was confused. Please describe each. The appropriate statistic to quote is the p-value (0.05 ??) with the number of observations used in each statistic, and one- or two-sided calculation, where there could be confusion. For example, a p–value of 0.10 would suggest to the reviewer that there was something worth further investigation. The point of maximum confusion for the reviewer was the discussion of drift. What number of samples was used? The eye sees that "independent" observations seem to occur often due to some rapidly changing condition: experimental error in one or both instruments, or rapid weather variation?

Reply:There is weather variation in ozone – see the first paper on Boulder Colorado – that is mostly day to day variation. Averaging over a week removes most of the weather variation.

Lowess(0.1), reference, explain "0.1)"?) Reply: Lowess(0.1) means that 10% of the total data were least squared averaged to form a smooth curve. It is the same as a "running average" except in the use of least squares instead of a linear average.

Lowess(1) would give a tradition linear least squares. This was explained in the original referenced paper.

The smoothed lines (which smoothing for Figure 3 as Figure 2. suggest that "weather" variation has a substantial impact on the smoothes and indeed the trends, especially in Figure 2. The smoothes for Figure 2 appear somewhat more convincing, but the uncertainty of 0.1% seems to be based on number of all samples rather than some partial contribution from "weather variability." One could guess a synoptic value of "five days per synoptic episode" and calculate a debatable approximate "number of samples" but the more appropriate value would be derived from a time series analysis which allowed for longer time-scales in that algorithm. In fact, there is enough excellent data here for most series to justify a more careful time-series analysis. For this publication, a disclaimer saying that "weather variability" could allow for a larger uncertainty in the apparent divergence is acceptable. In this case, "weather" is longer than one day but probably shorter than three years. Similar comments apply to the +/- 0.002 in Figure 1.

Reply: Since this paper is supposed to closely follow a 15 minute presentation at QOS, I will state that there is some weather variation and leave detailed statistical analysis for the future.

(minor points: explain acronym CCMI; Reply:CCMI is Chemistry–Climate Modelling Initiative

Reply: The acronyms for OMI and OMPS are given in the opening paragraph "Additional comparisons are made with satellite overpass data from OMI (Ozone Measuring Instrument on board the AURA spacecraft) and OMPS (Ozone Mapping Profiler on board the Suomi NPOESS satellite)."

perhaps OMI and OMPS are named on web pages, but could explained) This will be a nice addition to the description of stratospheric (and tropospheric) change and tropospheric change (TOAR). We may hope that the advent of many PANDORA instruments will add to a better discrimination of the variability and secular change of ozone as a

function of altitude. Minimal re-review is expected.

1. Does the paper address relevant scientific questions within the scope of AMT? Yes 2. Does the paper present novel concepts, ideas, tools, or data? Yes, Data 3. Are substantial conclusions reached? Yes, sufficient when they are qualified as noted 4. Are the scientific methods and assumptions valid and clearly outlined? Correctable. See notes above 5. Are the results sufficient to support the interpretations and conclusions? Ditto 6. Is the description of experiments and calculations sufficiently complete and precise to allow their reproduction by fellow scientists (traceability of results)? Ditto 7. Do the authors give proper credit to related work and clearly indicate their own new/original contribution? Yes 8. Does the title clearly reflect the contents of the paper? Yes 9. Does the abstract provide a concise and complete summary? Yes 10. Is the overall presentation well structured and clear? Yes 11. Is the language fluent and precise? Yes, but see 4. 12. Are mathematical formulae, symbols, abbreviations, and units correctly defined and used? Yes, minor additions needed for abbreviations, see above for e.g. "significant" and "Lowess(0.1)" 13. Should any parts of the paper (text, formulae, figures, tables) be clarified, reduced, combined, or eliminated? No 14. Are the number and quality of references appropriate? Yes 15. Is the amount and quality of supplementary material appropriate? Yes

---

## Referee Comment (RC2) · Anonymous Referee #2 · 20 Jul 2017

This paper presents comparisons among column ozone measurements at Boulder Colorado from two ground-based instruments and two satellite instruments. Daily data are analyzed for three years, and the focus of this short paper is to evaluate absolute differences among the measurement systems and quantify possible drifts (or trends) over the three years. I suppose the analysis is especially focused on evaluating the (relatively new) Pandora ozone measurements, although this is not explicitly stated. The results show small mean biases among the systems (+/- 1-2%), and excellent correlations for day-to-day and seasonal variability. The calculated difference trends show

small drifts (0.2 to 0.6 %/year) among the various measurements, and these drifts turn out to be statistically significant based on the results shown (Fig. 3). Note that the satellite comparisons suggest the largest drifts are associated with Dobson measurements. However, the authors downplay these significant trends and conclude that 'there is long-term stability in all four instruments'. In my opinion this summary statement needs to be better qualified in light of the significant trend results; I appreciate that the trends are derived from a short time record with arbitrary end points, with corresponding large uncertainties (the results look to be strongly influenced by the early 2014 data). But wouldn't drifts of magnitude ~6%/decade (as derived here) be troublesome if observed over a longer time record? I suggest that this detail needs some further discussion. Aside from this, I believe this short paper is a useful contribution to evaluating the Pandora ozone measurements, and is appropriate for AMT.

Minor comment: In line 40, Ozone Measuring Instrument should be Ozone Monitoring Instrument.

---

## Referee Comment (RC3) · Anonymous Referee #3 · 22 Jul 2017

General Comments: This paper gives a brief synopsis of comparisons for 3 years of Total Column Ozone (TCO) measurements from two ground-based (Dobson and Pandora) and two satellite-based (OMI and OMPS) platforms over Boulder, Colorado. The main objective is to analyze TCO differences between the instruments and find any trends (or drifts) over the short period. Since the Dobson instrument is usually a standard for TCO measurements, it would be worthwhile for the authors to mention this study as a validation effort of the Pandora, OMI and OMPS instruments (particularly those considered newer such as Pandora or OMPS). The comparisons presented give

valuable information, but further detail in the methodology of the statistics would provide more support for the interpretations the authors make. In addition, the discussion of the drifts found in the TCO measurements was missing any explanation for the results despite the highly correlated datasets. Drifts of 0.6%/year (or for the long term of 6%/decade) are not trivial, but appear to be minimized in the text.

Specific Comments: (1) Why was a Lowess fit (with 0.1) used versus another fit? If this analysis is related to what was presented in the Herman et al (2015) paper, this should be explicitly stated and any differences should also be pointed out. Is the fit used in Figure 2B the same as in Figure 3? If so, this should be stated. If not, an explanation is also needed. (2) The meaning of "significance" is not clear as written. What is used to test this? I think there is a level of assumption on the authors' part that we should know this, but some additional information would resolve any confusion. (3) After 2014, there is a noticeable separation between the TCO measurements between the Pandora and Dobson in Figure 2B. Do the authors have any explanation for this drift? The last statement of the summary including "long term stability of the four instruments" seems presumptive without any explanation for the observed trends. In my opinion, these results need to be characterized further to support that statement.

Minor comments: Line 40 – OMI and OMPS acronyms need to be corrected to Ozone Monitoring Instrument and Ozone Mapping Profiler Suite respectively. Line 88 – 'archived at WOUDC', missing "at". Line 102 – missing ";" to separate listed references.

After addressing the above concerns and clearing up some confusion in the results, I believe this paper would be appropriate for publication with AMT and provides useful evaluation of TCO observations over an extended time period.

1. Does the paper address relevant scientific questions within the scope of AMT? Yes 2. Does the paper present novel concepts, ideas, tools, or data? Yes - data 3. Are substantial conclusions reached? For the most part with some additional support

suggested in point 3 above. 4. Are the scientific methods and assumptions valid and clearly outlined? Yes except for the specific points 1 & 2 mentioned. 5. Are the results sufficient to support the interpretations and conclusions? Yes after point 3 is resolved. 6. Is the description of experiments and calculations sufficiently complete and precise to allow their reproduction by fellow scientists (traceability of results)? After specific points 1 & 2 are addressed. 7. Do the authors give proper credit to related work and clearly indicate their own new/original contribution? Yes 8. Does the title clearly reflect the contents of the paper? Yes 9. Does the abstract provide a concise and complete summary? Yes 10. Is the overall presentation well-structured and clear? Yes 11. Is the language fluent and precise? Yes 12. Are mathematical formulae, symbols, abbreviations, and units correctly defined and used? Same as 4. 13. Should any parts of the paper (text, formulae, figures, tables) be clarified, reduced, combined, or eliminated? No 14. Are the number and quality of references appropriate? Yes

---

## Author Response (AR1)

**Response:**

Since the manuscript was originally submitted, two significant changes have occurred. 1) NOAA updated the calibration of the Dobson#063 by applying the interim calibrations against the world standard Dobson#083. This changed all of the Dobson data slightly. 2) I investigated the influence of the endpoints on the percent difference time series and concluded that there was a significant effect on the slopes in some cases. Therefore, I de-seasonalized all of the percent difference time series as now described in the paper. All of the extensive changes in the paper are marked in green.

This is a good basic publication which just needs clarity and precision. Conclusions regarding trends in retrieved ozone need more modest standard error estimates, I believe The authors appear to make two assumptions:

(a) That "significance" means a 5% (? not stated) chance of Type 1 error (false acceptance) with a Gaussian distribution of errors.

The error estimates given are 1 standard deviation STD estimated from the least squares linear fit process. The error of the individual points was described in the previous paper as 1% with a precision of 0.1%. Most of the variation seen in the data is "natural" variation". It is clear that the error bars are large enough to make the statistical significance of some of the slopes marginal (see OMPS vs Pandora; 0.19 +/- 0.1). Some of the others are significant (see OMPS vs Dobson: -0.4 +/- 0.09) at the 2 STD level. P-values are now specified as are the number of points in each time series.

(b) That the "number of relevant samples" is the number of individual observations, apparently as averaged for 80 seconds for the PANDORA, the number of individual observations (averaged over 8 minutes, or once daily?) of observation recorded for the Dobson, and the number of days of observation (maximum once per day?) for OMI and OMPS.

Each Pandora data point is an average of 4000 measurements obtained during 30 seconds. All data for this study were clear-sky within the instrument's field of view based on the Dobson criteria for A-pair direct-sun clear sky. In addition, the Pandora data are averaged over a period of +/- 8 minutes surrounding the Dobson time of measurement (2 to 3 times per day). Pandora measurements are obtained every 80 seconds that means there were an additional 10 Pandora data points averaged together to compare to each Dobson measurement. The net averaging of Pandora is 40,000 (4x104) measurements for each comparison. The same procedure was used for comparisons with OMI and OMPS, where they measure once or twice per day over Boulder, Colorado.

For some comparisons, "data were selected for scenes that are clear-sky conditions as determined from the Dobson A pair" For all?

All Dobson vs Pandora, OMI, or OMPS scenes were clear-sky A-pair using the Dobson criterion

All Pandora vs OMI and OMPS were clear-sky or light clouds using a Pandora criterion measuring the noise between adjacent groups of measurements within the 4000 individual measurements that make up one Pandora data point.

**I added the following paragraph**

Each clear-sky PSI data point is an average of 2000 (early morning to evening SZAs) to 4000 (mid-day SZAs) measurements obtained during 20 seconds. All data for this study were clear-sky within the instrument's field of view based on the Dobson criteria for A-D-pair direct-sun clear sky. In addition, the PSI data are averaged over a period of +/- 8 minutes surrounding the Dobson time of measurement (2 to 3 times per day). Since PSI measurements are obtained every 80 seconds, there were an additional 10 PSI data points averaged together to compare to each Dobson, OMI, or OMPS measurement. The result is high signal to noise values for Pandora and high precision (0.1%). The same procedure using cloud-screened PSI data was used for comparisons with OMI and OMPS, where they measure once or twice per day over Boulder, Colorado. Some of the variations in the day to day ozone values are driven by changes in the local weather over Boulder, Colorado (see Fig. 14 in Herman et al., 2015), with weekly averages having much smaller variation.

How many days?

**Reply:**

The maximum number of days would be 1096. Not every day had a clear-sky observation and Pandora was not operational for some short periods. There are a significant number of days when OMI does not have an observation near Boulder.

Each of these numbers should be stated in the relevant context .

I now list the number of points in each time series in the graphs and summarize in a table.

| Table 1 Percent Difference Summary of Linear Fit Slopes and Mean Differences in Fig. 3 |                    |             |               |        |       |
|----------------------------------------------------------------------------------------|--------------------|-------------|---------------|--------|-------|
| Percent Diff(A,B)                                                                      | Slope (% per Year) | Probability | Mean (%)      | Points | Panel |
| Pan, Dob(BP)                                                                           | -0.2 ± 0.04        | P < 0.001   | -2.1 ± 1.6    | 2020   | А     |
| Pan, Dob(BDM)                                                                          | -0.2 ± 0.04        | P < 0.001   | -2.8 ± 1.6    | 2020   | В     |
| OMPS, Dob(BP)                                                                          | -0.09 ± 0.08       | P = 0.3     | -1.4 ± 2.1    | 854    | С     |
| OMI, Dob(BP)                                                                           | -0.18 ± 0.08       | P = 0.03    | -1.4 ± 1.9    | 654    | D     |
| OMPS, Pan                                                                              | -0.18 ± 0.098      | P = 0.06    | 0.96 ± 2.7    | 952    | Е     |
| OMI, Pan                                                                               | +0.18 ± 0.096      | P = 0.06    | $1.1 \pm 2.1$ | 624    | F     |

There are many statistics quoted where the reviewer was confused. Please describe each. The appropriate statistic to quote is the p-value (0.05 ??) with the number of observations used in each statistic, and one- or two-sided calculation, where there could be confusion. For example, a p-value of 0.10 would suggest to the reviewer that there was something worth further investigation. The point of maximum confusion for the reviewer was the discussion of drift. What number of samples was used? The eye sees that "independent" observations seem to occur often due to some rapidly changing condition: experimental error in one or both instruments, or rapid weather variation?

**Reply:**

**P-value is now included in the graphs**

Lowess(0.1), reference, explain "0.1)"?)

**Reply:**

Lowess(0.1) means that 10% of the total data were least squared averaged to form a smooth curve. It is the same as a "running average" except in the use of least squares instead of a linear average. Lowess(1) would give a tradition linear least squares. This was explained in the original referenced paper. Lowess(0.1) is roughly a 90-day low pass filter for this data set.

I added the sentence:

**The Lowess(f) procedure is based on local least squares fitting using low order polynomials applied to a specified fraction f of the data (Cleveland, 1979).**

The smoothed lines (which smoothing for Figure 3 as Figure 2. suggest that "weather" variation has a substantial impact on the smoothes and indeed the trends, especially in Figure 2. The smoothes for Figure 2 appear somewhat more convincing, but the uncertainty of 0.1% seems to be based on number of all samples rather than some partial contribution from "weather variability." One could guess a synoptic value of "five days per synoptic episode" and calculate a debatable approximate "number of samples" but the more appropriate value would be derived from a time series analysis which allowed for longer time-scales in that algorithm. In fact, there is enough excellent data here for most series to justify a more careful time-series analysis. For this publication, a disclaimer saying that "weather variability" could allow for a larger uncertainty in the apparent divergence is acceptable. In this case, "weather" is longer than one day but probably shorter than three years. Similar comments apply to the +/- 0.002 in Figure 1.

Reply There is weather variation in ozone – see the first paper on Boulder Colorado – that is mostly day to day variation. Averaging over a week would remove most of the weather variation. Deseasonalizing the percent difference time series removes any longer-term near periodic weather effects.

(minor points: explain acronym CCMI;

**CCMI is Chemistry–Climate Modelling Initiative**

The acronyms for OMI and OMPS are given in the opening paragraph

**"Additional comparisons are made with satellite overpass data from OMI (Ozone Measuring Instrument on board the AURA spacecraft) and OMPS (Ozone Mapping Profiler on board the Suomi NPOESS satellite)."**

perhaps OMI and OMPS are named on web pages, but could explained) This will be a nice addition to the description of stratospheric (and tropospheric) change and tropospheric change (TOAR). We may hope that the advent of many PANDORA instruments will add to a better discrimination of the variability and secular change of ozone as a function of altitude. Minimal re-review is expected. 1.

Does the paper address relevant scientific questions within the scope of AMT? Yes 2. Does the paper present novel concepts, ideas, tools, or data? Yes, Data 3.

Are substantial conclusions reached? Yes, sufficient when they are qualified as noted 4.

Are the scientific methods and assumptions valid and clearly outlined? Correctable. See notes above 5. Are the results sufficient to support the interpretations and conclusions? Ditto 6. Is the description of experiments and calculations sufficiently complete and precise to allow their reproduction by fellow scientists (traceability of results)? Ditto 7. Do the authors give proper credit to related work and clearly indicate their own new/original contribution? Yes 8. Does the title clearly reflect the contents of the

paper? Yes 9. Does the abstract provide a concise and complete summary? Yes 10. Is the overall presentation well structured and clear? Yes 11. Is the language fluent and precise? Yes, but see 4. 12. Are mathematical formulae, symbols, abbreviations, and units correctly defined and used? Yes, minor additions needed for abbreviations, see above for e.g. "significant" and "Lowess(0.1)" 13. Should any parts of the paper (text, formulae, figures, tables) be clarified, reduced, combined, or eliminated? No 14. Are the number and quality of references appropriate? Yes 15. Is the amount and quality of supplementary material appropriate? Yes

**Reply to Reviewer#2**

This paper presents comparisons among column ozone measurements at Boulder Colorado from two ground-based instruments and two satellite instruments. Daily data are analyzed for three years, and the focus of this short paper is to evaluate absolute differences among the measurement systems and quantify possible drifts (or trends) over the three years. I suppose the analysis is especially focused on evaluating the (relatively new) Pandora ozone measurements, although this is not explicitly stated.

**Response:**

Since the manuscript was originally submitted, two significant changes have occurred. 1) NOAA updated the calibration of the Dobson#063 by applying the interim calibrations against the world standard Dobson#083. This changed all of the Dobson data slightly. 2) I investigated the influence of the endpoints on the percent difference time series and concluded that there was a significant effect on the slopes in some cases. Therefore, I de-seasonalized all of the percent difference time series as now described in the paper. All of the extensive changes in the paper are marked in green.

**I have added the following to the Introduction:**

The recalibration of the Dobson and the de-seasonalization of the percent difference time series suggests that it is accurate to say in the introduction:

The results demonstrate the accuracy and stability of both the Dobson and PSI for retrieval of total column ozone.

The results show small mean biases among the systems (+/- 1-2%), and excellent correlations for day-today and seasonal variability. The calculated difference trends show small drifts (0.2 to 0.6 %/year) among the various measurements, and these drifts turn out to be statistically significant based on the results shown (Fig. 3).

1) Note that the satellite comparisons suggest the largest drifts are associated with Dobson measurements. However, the authors downplay these significant trends and conclude that 'there is long-term stability in all four instruments'. In my opinion this summary statement needs to be better qualified in light of the significant trend results; I appreciate that the trends are derived from a short time record with arbitrary end points, with corresponding large uncertainties (the results look to be strongly influenced by the early 2014 data). But wouldn't drifts of magnitude ~6%/decade (as derived here) be troublesome if observed over a longer time record? I suggest that this detail needs some further discussion.

**Response:**

The paragraphs discussing the comparison now reads"

Calculations for Pandora#034 (Panels E and F in Fig. 3) show marginally significant (p = 0.06) trends for Pandora#034 compared to OMPS (Panel E,  $-0.18 \pm 0.098$  % per year) and OMI (Panel F,  $+0.18 \pm 0.096$  % per year). If the Pandora#034 time series is extended into 2017 to minimize the effect of missing Pandora data in 2016, then the trends for Pandora compared to OMPS ( $-0.2 \pm 0.08$  % / Year p = 0.013) and OMI ( $0.15 \pm 0.076$  p=0.05) are significant, but not different from the shorter 2014 – 2016 period. The secular trends for the difference between Pandora#034 and Dobson#061 (-0.2% per year) are almost the same for both Dobson BP and BDM ozone absorption coefficients is small (0.042% per oC). This suggests that the stratospheric effective ozone temperature change is not a source for the small difference between Pandora#034 and Dobson#061.

Figure 4 shows that the TCO between Pandora#034 and Dobson#061 are highly correlated with 1:1 slope and the correlation coefficient  $r^2 = 0.97$  for the 3-year period 2014 to 2016. Similar correlation plots (Fig. 5) for Pandora#034 and Dobson#061 with OMI and OMPS also show very high correlations. The correlations in TCO are obtained after only temperature corrections to Pandora#034 and Dobson#061 using TE (TCO pairs similar to Fig. 2, panel A).

**And changed the Summary to read:**

Temperature corrected Pandora#034 and Dobson#061 differ by an average of 2.1% with Pandora using its standard retrieval BDM ozone absorption cross sections and Dobson using the recommended BP ozone absorption cross sections. Pandora compared to Dobson shows a small, but significant drift (-0.2  $\pm$  0.04 % per year. p < 0.001) for the 2014 – 2016 period. Comparisons of Pandora with OMI and OMPS are marginally significant drifts of 0.18 $\pm$ 0.1 and -0.18 $\pm$ 0.1 p=0.06 for 2014-2016, but are significant (0.15  $\pm$  0.076 % per year. p=0.05 and -0.2  $\pm$  0.08 % per vear. p = 0.013, respectively) if the period is extended to mid-2017 to minimize the effect of missing Pandora data during 2016. The small Pandora and Dobson trends compared to OMPS suggests that both instruments are stable. The conclusion is that the periodically calibrated Dobson#061 is able to detect smaller ozone trends than a Pandora instrument with no intermediate calibration during a 3-year period. The longer term trend for Dobson compared to OMPS for the 5.5-year period (2012 – June 2017) is -0.07  $\pm$  0.03 % per year, p = 0.047.

2) Aside from this, I believe this short paper is a useful contribution to evaluating the Pandora ozone measurements, and is appropriate for AMT.

Minor comment: In line 40, Ozone Measuring Instrument should be Ozone Monitoring Instrument. Corrected – Thank you

**Reply to Reviewer#2**

This paper presents comparisons among column ozone measurements at Boulder Colorado from two ground-based instruments and two satellite instruments. Daily data are analyzed for three years, and the focus of this short paper is to evaluate absolute differences among the measurement systems and quantify possible drifts (or trends) over the three years. I suppose the analysis is especially focused on evaluating the (relatively new) Pandora ozone measurements, although this is not explicitly stated.

**Response:**

Since the manuscript was originally submitted, two significant changes have occurred. 1) NOAA updated the calibration of the Dobson#063 by applying the interim calibrations against the world standard Dobson#083. This changed all of the Dobson data slightly. 2) I investigated the influence of the endpoints on the percent difference time series and concluded that there was a significant effect on the slopes in some cases. Therefore, I de-seasonalized all of the percent difference time series as now described in the paper. All of the extensive changes in the paper are marked in green.

**I have added the following to the Introduction:**

The recalibration of the Dobson and the de-seasonalization of the percent difference time series suggests that it is accurate to say in the introduction:

The results demonstrate the accuracy and stability of both the Dobson and PSI for retrieval of total column ozone.

The results show small mean biases among the systems (+/- 1-2%), and excellent correlations for day-today and seasonal variability. The calculated difference trends show small drifts (0.2 to 0.6 %/year) among the various measurements, and these drifts turn out to be statistically significant based on the results shown (Fig. 3).

1) Note that the satellite comparisons suggest the largest drifts are associated with Dobson measurements. However, the authors downplay these significant trends and conclude that 'there is long-term stability in all four instruments'. In my opinion this summary statement needs to be better qualified in light of the significant trend results; I appreciate that the trends are derived from a short time record with arbitrary end points, with corresponding large uncertainties (the results look to be strongly influenced by the early 2014 data). But wouldn't drifts of magnitude ~6%/decade (as derived here) be troublesome if observed over a longer time record? I suggest that this detail needs some further discussion.

**Response:**

The paragraphs discussing the comparison now reads"

Calculations for Pandora#034 (Panels E and F in Fig. 3) show marginally significant (p = 0.06) trends for Pandora#034 compared to OMPS (Panel E,  $-0.18 \pm 0.098$  % per year) and OMI (Panel F,  $+0.18 \pm 0.096$  % per year). If the Pandora#034 time series is extended into 2017 to minimize the effect of missing Pandora data in 2016, then the trends for Pandora compared to OMPS ( $-0.2 \pm 0.08$  % / Year p = 0.013) and OMI ( $0.15 \pm 0.076$  p=0.05) are significant, but not different from the shorter 2014 – 2016 period. The secular trends for the difference between Pandora#034 and Dobson#061 (-0.2% per year) are almost the same for both Dobson BP and BDM ozone absorption coefficients is small (0.042% per oC). This suggests that the stratospheric effective ozone temperature change is not a source for the small difference between Pandora#034 and Dobson#061.

Figure 4 shows that the TCO between Pandora#034 and Dobson#061 are highly correlated with 1:1 slope and the correlation coefficient  $r^2 = 0.97$  for the 3-year period 2014 to 2016. Similar correlation plots (Fig. 5) for Pandora#034 and Dobson#061 with OMI and OMPS also show very high correlations. The correlations in TCO are obtained after only temperature corrections to Pandora#034 and Dobson#061 using TE (TCO pairs similar to Fig. 2, panel A).

**And changed the Summary to read:**

Temperature corrected Pandora#034 and Dobson#061 differ by an average of 2.1% with Pandora using its standard retrieval BDM ozone absorption cross sections and Dobson using the recommended BP ozone absorption cross sections. Pandora compared to Dobson shows a small, but significant drift (-0.2  $\pm$  0.04 % per year. p < 0.001) for the 2014 – 2016 period. Comparisons of Pandora with OMI and OMPS are marginally significant drifts of 0.18 $\pm$ 0.1 and -0.18 $\pm$ 0.1 p=0.06 for 2014-2016, but are significant (0.15  $\pm$  0.076 % per year. p=0.05 and -0.2  $\pm$  0.08 % per vear. p = 0.013, respectively) if the period is extended to mid-2017 to minimize the effect of missing Pandora data during 2016. The small Pandora and Dobson trends compared to OMPS suggests that both instruments are stable. The conclusion is that the periodically calibrated Dobson#061 is able to detect smaller ozone trends than a Pandora instrument with no intermediate calibration during a 3-year period. The longer term trend for Dobson compared to OMPS for the 5.5-year period (2012 – June 2017) is -0.07  $\pm$  0.03 % per year, p = 0.047.

2) Aside from this, I believe this short paper is a useful contribution to evaluating the Pandora ozone measurements, and is appropriate for AMT.

Minor comment: In line 40, Ozone Measuring Instrument should be Ozone Monitoring Instrument. Corrected – Thank you

**Reply to Reviewer 3**

**Referee#3 Quad O3 Paper**

General Comments: This paper gives a brief synopsis of comparisons for 3 years of Total Column Ozone (TCO) measurements from two ground-based (Dobson and Pandora) and two satellite-based (OMI and OMPS) platforms over Boulder, Colorado. The main objective is to analyze TCO differences between the instruments and find any trends (or drifts) over the short period. Since the Dobson instrument is usually a standard for TCO measurements,

 it would be worthwhile for the authors to mention this study as a validation effort of the Pandora, OMI and OMPS instruments (particularly those considered newer such as Pandora or OMPS).

**Response:**

Since the manuscript was originally submitted, two significant changes have occurred. 1) NOAA updated the calibration of the Dobson#063 by applying the interim calibrations against the world standard Dobson#083. This changed all of the Dobson data slightly. 2) I investigated the influence of the endpoints on the percent difference time series and concluded that there was a significant effect on the slopes in some cases. Therefore, I de-seasonalized all of the percent difference time series as now described in the paper. All of the extensive changes in the paper are marked in green.

**I have added to the introduction**

The results demonstrate the accuracy and stability of both the Dobson and PSI for retrieval of total column ozone, and serves as a validation demonstration at one location for both the fairly new PSI and for satellite ozone data from OMI and OMPS.

2) The comparisons presented give valuable information, but further detail in the methodology of the statistics would provide more support for the interpretations the authors make. In addition, the discussion of the drifts found in the TCO measurements was missing any explanation for the results despite the highly correlated datasets. Drifts of 0.6%/year (or for the long term of 6%/decade) are not trivial, but appear to be minimized in the text.

**Response: The revised paper now has improved results due to the application of Dobson calibration and by the use of de-seasonalized percent difference (PD) time series. The drift of the Dobson relative to OMPS is now less than 1% per decade. The change was mostly an "end-point" effect of the PD time series.**

3) Specific Comments: (1) Why was a Lowess fit (with 0.1) used versus another fit? If this analysis is related to what was presented in the Herman et al (2015) paper, this should be explicitly stated and any differences should also be pointed out. Is the fit used in Figure 2B the same as in Figure 3? If so, this should be stated. If not, an explanation is also needed.

Response: I used a Lowess fit since it is the least squares equivalent of a running average that minimizes the effect of outliers. This is not to be confused with Loess(0.1) - I now give a reference for the Lowess algorithm.

The Lowess(0.1) is roughly a 90-day average, and as such acts as a "low-pass" filter on the data that can be used to derive a zero trend function needed to de-seasonalize the percent difference time series. Other functions could be used, but using the Lowess fit is one of the simplest starting points for deriving a zero-trend low-pass filter function.

**The caption to Figure 3 now reads**

Comparisons of Pandora(BDM) with Dobson(BP) retrieved ozone for Boulder, Colorado in percent differences of retrieved ozone and comparisons with OMI and OMPS. Slope = value of the linear least square fit,  $\pm N$  is 1 STD, and p is the probability (0 to 1) that the slope is statistically different from 0 relative to p = 0.05. The solid lines are a Lowess(0.1) fit and a linear least squares fit.

4) (2) The meaning of "significance" is not clear as written. What is used to test this? I think there is a level of assumption on the authors' part that we should know this, but some additional information would resolve any confusion.

**Response: I have added two criteria for significance 1) agreement to better than 2 standard deviations, and 2) the use of the p-value (probability of significance > 0.05).**

5) (3) After 2014, there is a noticeable separation between the TCO measurements between the Pandora and Dobson in Figure 2B. Do the authors have any explanation for this drift?

**No explanation. However, the net drift in the percent difference is now reasonably small (about 2% per decade)**

The last statement of the summary including "long term stability of the four instruments" seems presumptive without any explanation for the observed trends. In my opinion, these results need to be characterized further to support that statement.

Response: The revised time series analysis suggests that there is some drift in the OMI data, but that the other 3 instruments are stable (see Figure 3). OMI vs Dobson is statistically significant (p=0.03) at about -2% per decade while the drift with respect to OMPS (<1% per decade) is not statistically significant (p = 0.3). OMI vs Pan is about 2% per decade (marginally significant p=0.06) and OMPS vs Pan is about -2% per decade (p=0.06). If one assumes that the recalibrated Dobson is stable, then Pandora drifted downwards relative to the Dobson by a small amount, 2% per decade. 6) Minor comments: Line 40 – OMI and OMPS acronyms need to be corrected to Ozone Monitoring Instrument and Ozone Mapping Profiler Suite respectively.

**Response: Done**

7) Line 88 – 'archived at WOUDC', missing "at". Line 102 – missing ";" to separate listed references.

**Response: Fixed -**

After addressing the above concerns and clearing up some confusion in the results, I believe this paper would be appropriate for publication with AMT and provides useful evaluation of TCO observations over an extended time period. 1. Does the paper address relevant scientific questions within the scope of AMT? Yes 2. Does the paper present novel concepts, ideas, tools, or data? Yes - data 3. Are substantial conclusions reached? For the most part with some additional support suggested in point 3 above. 4. Are the scientific methods and assumptions valid and clearly outlined? Yes except for the specific points 1 & 2 mentioned. 5. Are the results sufficient to support the interpretations and conclusions? Yes after point 3 is resolved. 6. Is the description of experiments and calculations sufficiently complete and precise to allow their reproduction by fellow scientists (traceability of results)? After specific points 1 & 2 are addressed. 7. Do the authors give proper credit to related work and clearly indicate their own new/original contribution? Yes 8. Does the title clearly reflect the contents of the paper? Yes 9. Does the abstract provide a concise and complete summary? Yes 10. Is the overall presentation well-structured and clear? Yes 11. Is the language fluent and precise? Yes 12. Are mathematical formulae, symbols, abbreviations, and units correctly defined and used? Same as 4. 13. Should any parts of the paper (text, formulae, figures, tables) be clarified, reduced, combined, or eliminated? No 14. Are the number and quality of references appropriate? Yes Interactive comment on Atmos. Meas. Tech. Discuss., doi:10.5194/amt-2017-157, 2017.

**Marked Changes**

- 1 Ozone Comparison between Pandora #34, Dobson #061, OMI, and OMPS at Boulder
- 2 Colorado for the period December 2013 December 2016.
- 3
- 4 Jay. Herman1, Robert Evans4, Alexander Cede3, Nader Abuhassan1, Irina.
- 5 Petropavlovskikh2, Glenn McConville2, Koji Miyagawa5, and Brandon Noirot2
- 6

[revised manuscript text omitted]

| 265        | Debage $C = M = C = (1021) A$ shot exploring stress that the subscript of                      |
| 265        | atmospheric ozone. Proc. Phys. Soc. 13, 324, 339, 1031                                         |
| 268        | atmospherie ozone, 110e. 1 nys. 50e., 45, 524–557, 1751.                                       |
| 269        | Herman, J.R., R.D. Evans, A. Cede, N.K. Abuhassan, I. Petropavlovskikh, and G. McConville,     |
| 270        | Comparison of Ozone Retrievals from the Pandora Spectrometer System and Dobson                 |
| 271        | Spectrophotometer in Boulder Colorado, Atmos. Meas. Tech., 8, 3407–3418, 2015,                 |
| 272        | doi:10.5194/amt-8-3407-2015.                                                                   |
| 273        | Kombur W. D. P. D. Gross and P. K. Laonard (1980). Debson spectrophotometer 92: A              |
| ∠/4
275 | standard for total ozone measurements 1962–1987 I Geophys Res 94(D7) 9847–9861                 |
| 276        | doi:10.1029/JD094iD07p09847, 1989.                                                             |

[revised manuscript text omitted]

338 Figures339